# Pitfalls in the Diagnosis of Nodular Lymphocyte Predominant Hodgkin Lymphoma: Variant Patterns, Borderlines and Mimics

**DOI:** 10.3390/cancers13123021

**Published:** 2021-06-16

**Authors:** Sheren Younes, Rebecca B. Rojansky, Joshua R. Menke, Dita Gratzinger, Yasodha Natkunam

**Affiliations:** Department of Pathology, Stanford University School of Medicine, Stanford, CA 94305, USA; sfyounes@stanford.edu (S.Y.); rbloom@stanford.edu (R.B.R.); jmenke@stanford.edu (J.R.M.); ditag@stanford.edu (D.G.)

**Keywords:** LP cell, lymphocyte predominant, Hodgkin lymphoma, microenvironment, T-cell/histocyte-rich large B-cell lymphoma, diffuse large B-cell lymphoma

## Abstract

**Simple Summary:**

Nodular lymphocyte predominant Hodgkin lymphoma (NLPHL) is a rare lymphoma containing infrequent tumor cells (LP cells) in a background of non-neoplastic cells. Some cases of NLPHL can recur or progress to a more aggressive lymphoma, such as diffuse large B-cell lymphoma. Awareness of the different appearances of NLPHL and its overlap with other lymphomas are important for the appropriate diagnosis, classification and research. This article discusses the conceptual framework and guidelines for the diagnosis of NLPHL, and how NLPHL can be best separated from its mimics. Emerging data in the field point to genetic changes in LP cells that are shaped by immune mechanisms. In addition, non-neoplastic cells in the background of LP cells also appear to play an important role. Further investigation is necessary to fully understand the biology of NLPHL and personalize cancer care for patients affected by this lymphoma.

**Abstract:**

Nodular lymphocyte predominant Hodgkin lymphoma (NLPHL) represents approximately 5% of Hodgkin lymphoma and typically affects children and young adults. Although the overall prognosis is favorable, variant growth patterns in NLPHL correlate with disease recurrence and progression to T-cell/histiocyte-rich large B-cell lymphoma or frank diffuse large B-cell lymphoma (DLBCL). The diagnostic boundary between NLPHL and DLBCL can be difficult to discern, especially in the presence of variant histologies. Both diagnoses are established using morphology and immunophenotype and share similarities, including the infrequent large tumor B-cells and the lymphocyte and histiocyte-rich microenvironment. NLPHL also shows overlap with other lymphomas, particularly, classic Hodgkin lymphoma and T-cell lymphomas. Similarly, there is overlap with non-neoplastic conditions, such as the progressive transformation of germinal centers. Given the significant clinical differences among these entities, it is imperative that NLPHL and its variants are carefully separated from other lymphomas and their mimics. In this article, the characteristic features of NLPHL and its diagnostic boundaries and pitfalls are discussed. The current understanding of genetic features and immune microenvironment will be addressed, such that a framework to better understand biological behavior and customize patient care is provided.

## 1. Introduction

Nodular lymphocyte predominant Hodgkin lymphoma (NLPHL) is a rare tumor with an annual incidence of 0.1–0.2 per 100,000. Although it affects patients of all ages, it has a marked predilection for children and young adults and shows a 3:1 male predominance [1,2,3,4]. Similar to classic Hodgkin lymphoma (CHL), NLPHL contains very scarce tumor cells called lymphocyte-predominant (LP) cells that are ensconced in an inflammatory tumor microenvironment (TME) that is enriched for lymphocytes and histiocytes. Clinically, the vast majority of NLPHL patients (>75%) present with stage I and -II diseases and typically involve peripheral lymph nodes without extranodal, mediastinal or hepatosplenic involvement. Approximately 20% of patients, however, exhibit bulky disease and/or hepatosplenomegaly and associated systemic B symptoms. Up to 20–30% of patients with NLPHL experience recurrences and/or progression; nevertheless, the prognosis is favorable, with a 10-year overall survival of >80% [4,5,6,7,8,9,10,11,12].

Approximately 3–17% of NLPHL show progression to large B-cell lymphoma, which may be either T-cell/histiocyte-rich large B-cell lymphoma (THRLBCL) or diffuse large B-cell lymphoma (DLBCL) [13,14,15,16,17,18,19,20,21,22,23,24,25,26]. Patients with splenic, subdiaphragmatic or bulky disease have a higher risk of progression. Unlike NLPHL, de novo THRLBCL is associated with an aggressive clinical course, although a less-aggressive clinical behavior is recognized at least in some cases that are transformed from NLPHL [17,18,19,20,21,22]. Defining the robust criteria for the separation of NLPHL from THRLBCL can be challenging in some cases where there is histologic and immunophenotypic overlap. Given the clinical and prognostic differences between NLPHL and THRLBCL, it is imperative that they are distinguished from one another.

We previously identified variant immunoarchitectural growth patterns in NLPHL that correlated with disease recurrence and progression [27]. The prognostic relevance of these variant patterns was confirmed in independent adult and pediatric patient cohorts [28,29,30], and has been incorporated into the World Health Organization (WHO) classification of NLPHL and by the National Comprehensive Cancer Network (NCCN) clinical practice guidelines [4,10]. The underlying changes responsible for the altered growth characteristics are poorly understood. One of the sparsely explored areas is the spatial configuration and composition of the TME. This TME is composed of lymphocytes, histiocytes and dendritic cells, which vastly outnumber tumor B-cells. However, conventional immunohistochemistry and flow cytometry have not been sufficiently informative in understanding these spatial relationships and how they influence tumor behaviors. Similarly, high-throughput molecular methods that are informative in bulk tumor samples have also not fully captured the subtleties embedded in the histologically defined patterns. Therefore, the spatial configuration of the TME and its contribution to lymphoma evolution needs further study.

## 2. Histologic and Immunophenotypic Diagnosis of NLPHL

NLPHL is characterized by the nodules of small lymphoid cells associated with follicular dendritic cell (FDC) meshworks and scattered large neoplastic LP cells that reside within and outside nodules (Figure 1A–C). At least some nodularity is necessary to render a diagnosis of NLPHL. Prominent sclerosis is found in over 20% of cases [27], and can range from fine reticular fibrosis to fibrous bands; sclerosis is more prevalent in recurrent biopsies. (Figure 1D). Epithelioid histiocytes are also abundant in the TME, and in some cases, they surround lymphoid follicles in clusters or form epithelioid granulomas (Figure 1E). Entrapped reactive follicles may also be present. The number of scattered LP cells are variable and range from 1–5% of the tumor mass; rare nodules may show an increased number or confluence of LP cells (Figure 1F).

The NLPHL TME is rich in lymphocytes and histiocytes, with a predominance of small, mature, non-neoplastic B cells. In addition, there are admixed T cells with a subset of T-follicular helper (TFH) cells that encircle LP cells in a ring or rosette-like formation. Awareness of the histologic features is helpful and enables an informed selection of ancillary studies to distinguish NLPHL from its morphologic mimics. The diagnosis of NLPHL therefore requires immunohistochemistry to outline the architectural configuration as well as the phenotypes of the tumor and TME cell types. Precise criteria regarding the number of nodules, number of LP cells within the nodules and numbers and cell types in the TME are largely undefined, and therefore, the NLPHL disease definition remains descriptive. 

The typical immunophenotypic features of NLPHL are shown in Figure 2 and summarized in Table 1. Commensurate with the germinal center (GC) B-cell origin of LP cells [31], they express pan B-cell markers and rearranged immunoglobulins with kappa or lambda light-chain expression (Figure 2A–C). GC B-cell markers are expressed with the exception of CD10. LP cells may rarely show a loss of B-cell markers, particularly CD20 [32], and also express CD30, and less frequently, CD15 [27,33,34,35,36]. The robust expression of B-cell transcription factors PAX5, OCT2 and BOB1 in the majority of NLPHL aids in the separation from CHL [35,36,37] (Figure 2D–F). Additionally, MEF2B positivity and a lack of nuclear STAT6 have been shown to distinguish NLPHL from CHL [38,39].

Follicular dendritic cell markers such as CD21 and CD23 are helpful to highlight the FDC meshworks associated with the B-cell nodules within which the LP cells reside (Figure 2G). In some cases, FDC markers are indispensable to bring out a nodular configuration, which may be difficult to appreciate on hematoxylin and eosin (H&E) stains, particularly when there is limited tissue architecture, such as in core needle biopsies or when variant growth patterns are present. Furthermore, the detection of TFH rosettes surrounding LP cells, which is best exemplified by PD1 immunohistochemistry, is a valuable feature in the diagnosis of NLPHL (Figure 2H,I) [45,46]. Approximately 27–50% of NLPHL show IgD expression on LP cells, which defines a particular clinicopathologic subtype of NLPHL with a strong male predominance, extrafollicular LP cells, propensity for recurrence and a potential pathogenetic mechanism related to the *Moraxella* species [40,41,42].

The immunophenotypic panel used in pathology practice may vary depending on diagnostic expertise and availability of markers. A tiered approach to ancillary immunostains with the recommended, desirable and optional categories of markers is provided in Table 2. Additionally included are the relevant stains that should be considered when specific differential diagnostic considerations arise. At the current time, flow cytometry, cytogenetics/ flourescent in situ hybridization (FISH) and molecular studies are not routinely employed for the diagnosis of NLPHL, although these strategies are being increasingly explored for clinical adoption in the future [43,44,52,53,54,55].

## 3. Six Immunoarchitectural Growth Patterns of NLPHL

Our description of the six patterns of NLPHL were defined by a combination of histologic and immunophenotypic features [27]. Schematic representations of these patterns are summarized in Figure 3. The first two patterns are exemplified by classic nodules (pattern A) or serpiginous/interconnected nodules (pattern B); the key characteristic of both A and B patterns is that LP cells predominantly reside within the confines of small B-cell-rich nodules. The next two patterns maintain a nodular growth associated with FDC meshworks although, there are a significant number of extranodular LP cells (pattern C) or nodules that are rich in small T cells (pattern D). The last two patterns show diffuse growths with a TME that is enriched either for T cells and histiocytes (pattern E) or small B cells (pattern F). A comparison of the six patterns with corresponding CD20-stained images of representative biopsies are shown in Figure 4.

Several clinically relevant insights were gleaned from the Fan patterns: (1) the classic nodular pattern A was more likely to present as a pure pattern (39%), whereas a mixture of two patterns was seen in 25% and three or more patterns in 31% of biopsies; (2) extranodular LP cells were associated with a propensity for diffuse growth and loss of FDC meshworks in serial biopsies over time; (3) the presence of a diffuse THRLBCL-like pattern E was more common in patients with recurrent disease and was an independent predictor of recurrence; (4) in diffuse patterns E and F, the presence of nodules elsewhere in the same biopsy was essential for inclusion in NLPHL. Therefore, a diagnosis of a pure diffuse THRLBCL-like pattern is not possible because that would constitute a diagnosis of THRLBCL and, (5) in rare patients, DLBCL preceded the occurrence of NLPHL, which further underscores the close biologic relationship between these entities.

The clinical impact of the six patterns were further explored by Hartmann and colleagues in 423 patients encompassing nine clinical trials performed by the German Hodgkin Study Group [28]. In this cohort, the majority of the patients (75%) had patterns A and B, whereas 25% of patients exhibited patterns C–F, also called variant patterns. The presence of variant patterns was found to be an independent prognostic factor associated with advanced disease and higher relapse rates. In addition, variant histology at the initial diagnosis was found to persist at a recurrence in the majority of patients and variant histology at initial diagnosis as well as at a relapse correlated with a shorter time to progression of the disease [29]. Similarly, variant growth patterns of NLPHL were found to be associated with a higher stage and lower complete response and more frequent relapse rates in the pediatric age group [30].

The recognition of different growth patterns and their clinical implications raise several considerations for pathologists in the diagnosis of NLPHL. Most importantly, they underscore the need for careful histologic and immunophenotypic evaluations, including the overall architecture with nodular and diffuse components, the number and distribution of LP cells, and the cell composition of the TME throughout the biopsy. This close assessment may identify features that herald the progression or recurrence of disease or, conversely, detect a single nodule in an otherwise diffuse growth pattern and avert a diagnosis of large B-cell lymphoma. Furthermore, the pathologists’ awareness of the immunophenotypic variation that is acceptable for the diagnosis of NLPHL is necessary to trigger additional immunophenotypic or molecular workups if unusual phenotypes are found. 

## 4. Progression of NLPHL to THRLBCL and DLBCL 

The clinical course of NLPHL is typically indolent; however, a subset of patients suffers multiple recurrences, as well as progression to a large B-cell lymphoma, which is reported to occur in 3–17% of patients [13,14,15,16,17,18,19,20,21,22,23,24,25,26]. Progression to a large B-cell lymphoma involves a characteristic loss of nodular architecture and FDC meshworks, and the large B cells become scattered individually (characteristic of THRLBCL) or form clusters and sheets (characteristic of DLBCL). The transition also involves changes in the TME, which typically progress from a B-cell-rich microenvironment to one that is rich in T cells and histiocytes [19]. Individual cases, however, show wide variations in LP cell numbers and confluence, as well as the TME composition, making reliable predictions regarding the biologic behavior challenging.

THRLBCL may arise de novo or progress from NLPHL. De novo THRLBCL is a rare entity that presents in middle-aged patients, exhibits a male predominance and is associated with systemic involvement, including fever, malaise, splenomegaly and/or hepatomegaly. More than 50% of patients present with advanced stage (medium- to high- risk) disease, which follows an aggressive course that is often refractory to chemotherapy [13,14,15,16,17,18,19,20,21,22,23,24,25,26]. Heterogeneity in clinical behavior has also been recognized, especially the less-aggressive disease course reported in patients who likely progressed from a known or previously unrecognized NLPHL. Apart from the international prognostic index (IPI) which is highly correlated with a prognosis in THRLBCL, there is a dearth of robust immunophenotypic or molecular markers to stratify the risk or to separate de novo THRLBCL from those progressed from NLPHL. 

## 5. Diagnostic Mimics and Overlaps

NLPHL and its variant growth patterns may morphologically mimic other lymphomas, including THRLBCL/DLBCL, CHL and mature T-cell lymphomas such as peripheral T-cell lymphoma (PTCL and NOS) or angioimmunoblastic T-cell lymphoma (AITL). NLPHL can also overlap with reactive entities such as the progressive transformation of germinal centers (PTGC). 

### 5.1. Overlap with THRLBCL/DLBCL

The spectrum of NLPHL and THRLBCL/DLBCL is increasingly recognized as a true biologic continuum. In the last two decades, the overlap in clinical, as well as histologic, features have been well-studied by many investigators. More recently, the overlap in TME cell types and composition, gene expression signatures and mutational profiles, as revealed by next-generation sequencing, have also been elucidated [55,56,57,58,59,60,61,62]. These investigations and awareness of the biologic continuum have been instrumental in refining the conceptual framework of NLPHL-THRLBCL/DLBCL and serve as a guide for clinical practice. 

Typical histologic and immunophenotypic features of THRLBCL include a diffuse architecture with limited number of atypical large B cells associated with abundant T cells and histiocytes without an associated nodular component (Figure 5A–C). The atypical large B cells in THRLBCL account for approximately 10% or less of the tumor mass, although there is wide variability and lack of well-defined numbers or standardized measures to quantify tumor cells in this context. The THRLBCL-like pattern of NLPHL (pattern E) shows a significant overlap with THRLBCL. In addition, there is a loss of PD1-rings in variant growth patterns, which is a helpful feature to distinguish NLPHL from THRLBCL [45,46]. The loss of PD1 rings generally mirrors a progressively more diffuse growth pattern and further confounds the diagnostic distinction between NLPHL and THRLBCL. PD1 rings are a notoriously unreliable feature, as they can be seen in THRLBCL and CHL as well as reactive entities such as viral infections, including infectious mononucleosis [46,63]. A representative image of THRLBCL with PD1 rings surrounding large B cells, as well as histiocytes is shown in Figure 5D.

Given the close relationship of NLPHL and THRLBCL, there are two very important considerations at the time of diagnosis: (1) a careful pathology assessment to exclude the presence of NLPHL, especially the THRLBCL-like/pattern E, and (2) clinical staging to assess the distribution and burden of the disease. Similarly, at the time of suspected recurrence or progression, re-biopsy and re-staging are important for confirmation of the diagnosis. These pre-emptive actions will serve to avoid an over- or underdiagnosis and enable the customization of patient care.

### 5.2. Overlap with Classic Hodgkin Lymphoma

Another difficult boundary exists between NLPHL and CHL, particularly lymphocyte-rich CHL (LR-CHL; Figure 5E–H). Both entities exhibit scattered atypical large cells in a background that is enriched for small lymphocytes. Their immunophenotypes may also overlap with the large cells expressing CD20 and CD30 [27,33,34,35,36]. CD15 positivity, however, is infrequent (2–7%) in NLPHL [27,33,34,35,36]. Unlike the abrogated B-cell program in CHL due to defective B-cell receptor-mediated transcription and signaling, NLPHL exhibits an intact B-cell program with a robust expression of pan-B markers and B-cell transcription factors [31,35,36,37]. In occasional NLPHL cases, however, there has been a partial or complete loss of B-cell transcription factors, particularly PAX5 (Figure 5G). In such cases, additional B-cell markers, including CD79a, BCL6, OCT2 and BOB1, may be necessary to further substantiate the B lineage and confirm the diagnosis of NLPHL. 

Several new markers have been described recently that are useful adjuncts in the separation of NLPHL from CHL. MEF2B is positive in LP cells of NLPHL and negative in CHL [38]. STAT6 shows a nuclear positivity in CHL, whereas a nuclear positivity is absent in LP cells, although weak cytoplasmic staining can be detected in up to 74% of cases [39]. Although less commonly employed in a clinical setting, FISH for BCL6 is positive in NLPHL but not in CHL [44,53]. PD-L1 immunohistochemistry and FISH for *9p24.1* aberrations are a defining feature of CHL [64,65] and in select other lymphomas, including THRLBCL [62]. The utility of PD-L1 FISH in NLPHL and its variants await rigorous investigation and validation and, if confirmed, could become a valuable aid in the clinical setting.

The Epstein–Barr virus can be detected in tissue biopsies of rare NLPHL and has been reported in up to 5% of cases [49,50]. EBER in situ hybridization is the most sensitive method for demonstrating EBV within LP cells (Figure 5H). EBV may also be detected in bystander cells in the TME and has been postulated to represent host immune dysregulation. Furthermore, EBV can be expressed in immunoblasts in the TME, which can easily be confused for Hodgkin cells, particularly in small biopsy samples with limited architecture. The role of EBV in NLPHL pathogenesis is less well-understood compared to EBV’s role in CHL, where EBV-positive cases have been shown to harbor a higher mutational burden than those without EBV [66]. These findings have led to the hypothesis that EBV may function as a transient trigger of lymphomagenesis in CHL by providing a permissive environment for aberrant GC B-cell clonal evolution. Direct evidence of the contribution of this mechanism in NLPHL has not been confirmed and awaits further investigation. In the diagnostic setting, the detection of EBV in LP cells of NLPHL further complicates its separation from EBV+ CHL, especially since EBV typically upregulates the expression of CD30 and PDL1 [47,48]. 

The significant overlap in the morphology and immunophenotype between NLPHL and CHL raises distinct diagnostic challenges, particularly because of the important differences in the biologic behavior, prognosis and, therefore, the disparate treatment strategies that are used for each. Given the rarity of the tumor cell in both entities, additional ancillary studies, including molecular diagnostics, are not likely to be informative. A multidisciplinary approach with a correlation of clinical, pathologic and imaging/staging is essential in such cases to select the most appropriate management for the patient. Although not formally recognized as a borderline category (so-called “grey-zone”) by the WHO classification, the overlap between NLPHL and CHL may deserve to be conceptualized in a similar manner.

### 5.3. Overlap with Mature T-Cell Lymphomas

Although uncommon, a subset of NLPHL has an increased T-cell content with the TME that can mimic a T-cell lymphoma [67,68,69]. Since these T cells typically express TFH markers such as PD1, ICOS, BCL6 and CXCL13, they can be mistaken for angioimmunoblastic T-cell lymphoma or a peripheral T-cell lymphoma with a TFH phenotype. An increased T-cell content is a feature of NLPHL with variant growth patterns D and E, as well as a feature of the progression to THRLBCL. Loss or aberrant T-cell marker expressions by flow cytometry or immunohistochemistry, as well as molecular T-cell receptor gene rearrangements are very helpful in separating NLPHL/THRLBCL from a T-cell lymphoma. If confluent tumor cell sheets are present, a combination of B- and T-cell receptor gene rearrangements have the highest likelihood of providing definitive results. A correlation with the clinical features and staging are additional important metrics to take into consideration in ruling out a T-cell lymphoma. 

### 5.4. Overlap with Progressive Transformation of Germinal Centers

PTGC is a self-limiting lymphadenopathy of unknown etiology that can present at any age from early childhood into older adulthood. Although not considered a precondition, the synchronous or metachronous co-occurrence of PTGC and NLPHL is well-described, although rare (3.5% in one study) [70,71]. We previously showed that there are several architectural changes that occur during the dismantling of germinal centers in PTGC with and without associated NLPHL [72]. Overlapping features include enlarged and atypical lymphoid nodules with disruption of the normal germinal center morphology, atypical mantle zones, FDC meshworks that show a range of mild-to-severe disruptions to prominent expansions, the atypical localization of normal GC cell types, including PD1-positive TFH cells, and a cell milieu with an identical GC B-cell phenotype (Figure 5I–L). 

The key feature that permits the separation of PTGC from NLPHL is the lack of LP cells in PTGC, along with the lack of PD1 rings. Identifying LP cells, however, may be confounded by prominent centroblasts, follicular dendritic cells and histiocytes, which may falsely impart a resemblance to LP cells. In such cases, looking for extranodular LP cells can provide an important clue. Furthermore, while both entities show a range of misshapen nodular structures, NLPHL nodules frequently exhibit ragged outlines in comparison to PTGC nodules, as best visualized on an IgD stain. A BCL2 stain can also provide helpful information, although the BCL2 expression in paracortical T-cell zones makes the visualization of a clear separation from mantle zone B cells less optimal. Increased IgG4-positive cells have been described in association with PTGC but not NLPHL, and therefore, IgG4 is another useful marker to separate PTGC from NLPHL [51]. Given the often-subtle differences between PTGC and NLPHL, awareness is important to avoid misdiagnosis.

### 5.5. Diagnostic Pitfalls in Core Needle Biopsies

The diagnosis of NLPHL in core needle biopsies (CNB) is particularly challenging given the limited tissue architecture. The optimal approach to a diagnosis will depend on whether CNB is performed for the initial diagnosis versus an evaluation of recurrence and/or progression. In our recent study comparing fine-needle aspirations (FNA) with CNB versus excisional or incisional surgical biopsies, we found that a surgical biopsy remains the gold standard for the initial diagnosis of NLPHL [73]. The differential diagnosis in the initial diagnostic setting of NLPHL is broad and, most importantly, would include CHL. The choice of immunohistochemistry should therefore include the following markers: CD20, CD30, CD3 and PAX5 or OCT2, CD21 and EBV (EBER in situ hybridization). The results of this initial panel could then be used to guide further workups as necessary (Table 2 and Figure 6). For example, if CD20 is absent (1–3% of NLPHL) or focal/weak (6%), as we recently described [32], additional B-cell markers (CD79a, PAX5, BOB1 and CD19) can be performed to confirm a B-lineage. The inclusion of CD15 would be useful to confirm a diagnosis of CHL with the caveat that CD15 expression in LP cells has infrequently (positive in 2–7% of NLPHL) been reported [27,34,35,36]. If the initial FNA/CNB is suggestive of NLPHL, a surgical (excisional or incisional) biopsy should be performed for a definitive diagnosis [73].

In contrast, for CNB in patients with a confirmed history of NLPHL and under the appropriate clinical setting, the focus should be on ruling out the recurrence and/or progression to THRLBCL/DLBCL. In this context, the choice of markers that would be most useful includes OCT2, CD21, IgD, PD1 and CD3. OCT2 is almost always overexpressed in LP cells and is superior to CD20, which may be negative due to a prior rituximab therapy. OCT2 staining should also be interpreted with caution, since reactive immunoblastic proliferations can show a bright expression of OCT2. If in doubt, additional B-cell markers such as CD79a, PAX5 and BOB1 should be obtained for confirmation. In CNB, FDC markers (CD21 or CD23) provide valuable information regarding the presence or absence of nodularity. IgD is useful in a similar architectural context and, in addition, to identifing the IgD-positive subtype of NLPHL, which is prone to more frequent recurrences [40,41,42]. Caution should be exercised in the interpretation of PD1 in needle core biopsies; although the presence of PD1-positive rings is helpful, the separation of the THRLBCL-like pattern of NLPHL from the progression to THRLBCL/DLBCL can be particularly challenging in CNB. In addition, PD1 can show extensive staining in background cells within the TME, which may be difficult to interpret in the context of limited tissue architecture [65]. CNB is capable of providing a definitive diagnosis in scenarios where a distinct nodular architecture is present with the associated FDC meshworks or if diffuse sheets of atypical large B cells are present, and this finding is aligned with the clinical evidence of disease progression. 

## 6. Emerging Concepts in NLPHL 

Recent investigations into the genetic landscape of the NLPHL-THRLBCL/DLBCL have provided new insights into the pathogenesis and progression of this lymphoma spectrum. NLPHL arises from clonally expanded, highly mutated GC B cells with ongoing somatic hypermutations [31]. Using immunoglobulin heavy-chain next-generation sequencing (NGS), Paschold and colleagues recently showed that, in recurrent NLPHL, LP *IGH* rearrangements were identical at diagnosis and relapse, whereas tNLPHL showed intraclonal diversification with clonal evolution. These studies suggest the potential role of an antigenic drive in NLPHL transformation and the utility of using targeted NGS to predict the high transformation risk [74]. Mutational profiling by NGS has further shown that LP cells harbor recurrent genetic alterations and deregulated signaling pathways involving *SGK1, DUSP2* and *JUNB* in NLPHL and *SGK1, DUSP2, JUNB, SOCS1* and *CREBBP* in THRLBCL. In addition, transformed NLPHL (tNLPHL) has distinct genetic profiles and harbors a comparable number of genomic alterations to that of de novo DLBCL [55,60,61,75,76,77,78]. Frequent mutations in the PI3K and NF-κB signaling pathways and epigenetic modifiers have also been demonstrated with a similar mutational profile to that of germinal center (GC) B-cell-derived DLBCL [61]. In addition, tNLPHL harbors frequent mutations in *TET2*, *JUNB* and *NOTCH2*, which are similar to the less-frequent “BN2-mutational signature” of DLBCL that typifies chronic active BCR stimulation [77]. Song and colleagues further showed the resemblance in the mutational profiles between tNLPHL and tFL and raised the important consideration that the loss/changes in the TME during progression to DLBCL likely underlie this similarity [61]. This observation emphasizes the important contribution of the TME to NLPHL pathogenesis and progression. Recent advances in multiplexing and computational technologies have allowed the precise measurements of spatially resolved configurations pertinent to the tissue architecture and cell composition, as well as to the metabolic tumor cell volume [79,80,81,82,83], thereby providing exciting new avenues for future research. 

Approximately 25% of NLPHL exhibit IgD-positive LP cells and demonstrate specific clinicopathologic and genetic features. IgD-positive NLPHL have a striking male predominance, extrafollicular LP cells with variant patterns C and E and a higher likelihood of transformation into DLBCL [40,41,42]. An elegant model of *Moraxella*-induced lymphomagenesis has recently revealed infection/inflammation as a potential trigger for the development of IgD-positive NLPHL [42]. The immune-related pathogenetic mechanism shows overlap with disorders of immune dysregulation and offers the opportunity to explore novel therapeutic approaches with sustained remission and less toxicity. In addition, TFH rosettes were found to engage the major histocompatibility complex class II beta chain expressed by LP cells through the T-cell receptor alpha chain in an immunologic synapse [84]. This finding not only provides a functional correlate to the observed PD1 configuration but also suggests a close bidirectional relationship between the LP cells and the TFH cells that encircle them within the microenvironment. 

Familial cases of NLPHL bring additional insights to understanding the pathogenesis of NLPHL. In a population-based Finnish registry study, the familial risk was found to be much higher in NLPHL with a standard incidence ratio of 19 compared with 5.3 in CHL and 1.9 in non-Hodgkin lymphoma [85]. Exome sequencing identified truncating germline mutations in the ataxia-telangiectasia locus (*NPAT* gene) in a family cluster of four cousins [86]. NLPHL has also been reported in patients with Hermansky-Pudlak type 2 syndrome, a rare autosomal recessive primary immune deficiency associated with functional defects in cytotoxic activity in NK cells and increased susceptibility to infections and lymphoma [87]. Additional cases associated with autoimmune lymphoproliferative syndrome (ALPS) with germline *FAS* mutation, germline *TP53* in Li-Fraumeni syndrome, gain-of-function *STAT1* mutation and *TET2* haploinsufficiency have also been reported [88,89,90]. These rare but informative genetic associations suggest immune dysregulation as a potential mechanism in NLPHL pathogenesis.

The insights gleaned from recent investigations have served to advance our knowledge of NLPHL biology. Large validation studies with adequate numbers of rare variants and borderline cases are necessary to better define the boundaries surrounding NLPHL. It is anticipated that state-of-the-art genomic and spatial profiling technologies will propel discoveries forward and guide the refinement of diagnostic criteria and therapeutic approaches.

## 7. Conclusions

NLPHL is a typically indolent lymphoma with a propensity for recurrence and/or progression. Its rarity, together with its spatial configuration of sparse tumor cells surrounded by a complex TME, present challenges for its study, as well as its diagnosis. Variant growth patterns are associated with the recurrence and progression of diseases and should be recognized and specified in pathology reports. These patterns also provide a conceptual framework for the diagnosis and further study of the biologic continuum between NLPHL and THRLBCL/DLBCL, as well as CHL. A multidisciplinary approach is beneficial to optimize the diagnosis and management of patients with this spectrum of lymphomas.

## Figures and Tables

**Figure 1 cancers-13-03021-f001:**
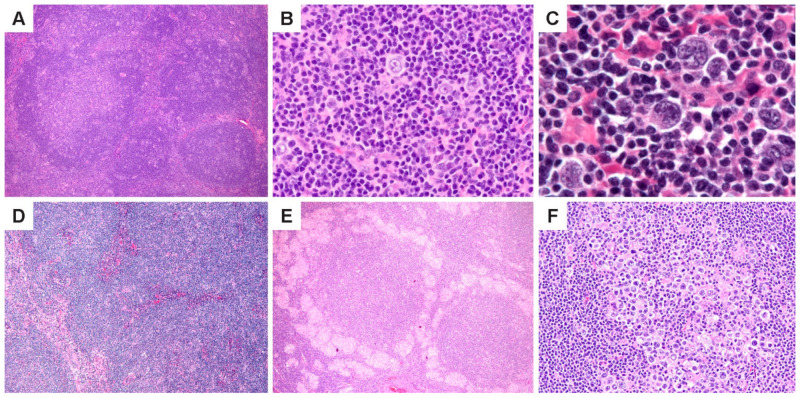
Histologic features of NLPHL. (**A**) Lymph node effaced by a proliferation of classic lymphoid nodules. (**B**,**C**), Scattered LP cells in a microenvironment rich in small lymphocytes and histiocytes. (**D**) Nodules interspersed by bands of fibrosis. (**E**) Nodules surrounded by epithelioid histiocytes. (**F**) A nodule with increased LP cells. [Original magnifications: **A**,**D**,**E** ×60; **B**,**F** ×150; **C** ×600].

**Figure 2 cancers-13-03021-f002:**
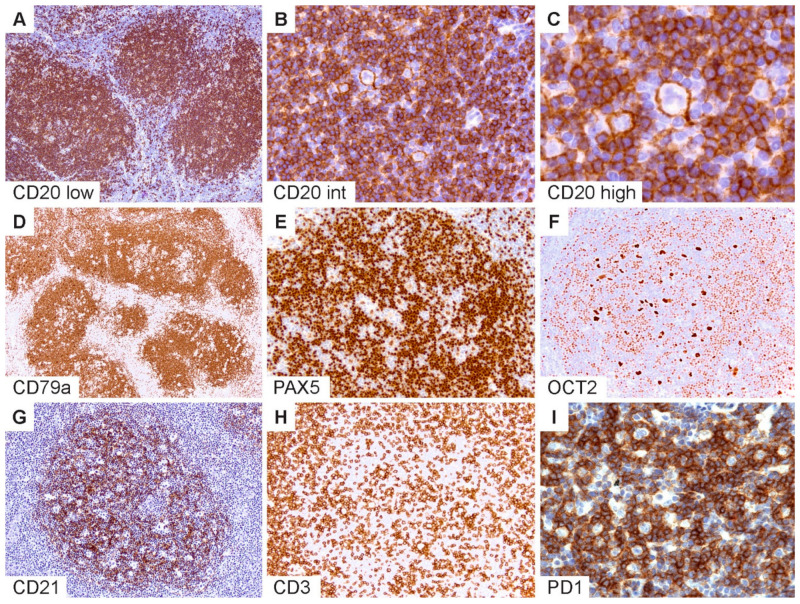
Immunophenotypic features of NLPHL. (**A**) CD20 highlights numerous classic lymphoid nodules. (**B**,**C**) Scattered LP cells reside within nodules containing a microenvironment rich in small B-lymphocytes. (**D**) CD79a, (**E**) PAX5 and (**F**) OCT2 show staining of LP cells and the surrounding small B cells. (**G**) CD21 defines an intact follicular dendritic cell meshwork within an NLPHL nodule. (**H**) CD3 stains background T cells with occasional ring formations and (**I**) prominent PD1-positive rings surrounding LP cells. [Original magnifications: **A**,**D**,**G**,**H** ×60; **B**,**E**,**F**,**I** ×150; **C** ×600].

**Figure 3 cancers-13-03021-f003:**
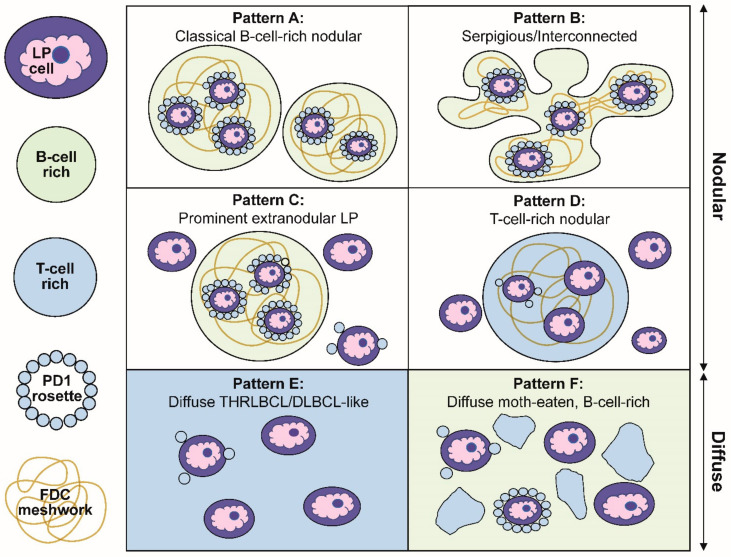
Schematic representation of the six immunoarchitectural patterns of NLPHL. (**A**) Classical B-cell-rich noduless and (**B**) serpiginous/interconnected nodules define the typical patterns of NLPHL, with most LP cells residing within the nodules associated with FDC meshworks. (**C**) Prominent extranodular LP cells define pattern C, whereas (**D**) T-cell-rich nodules define pattern D. (**E**) The diffuse THRLBCL/DLBCL-like pattern and (**F**) diffuse moth-eaten B-cell-rich pattern show diffuse growths unassociated with the FDC meshworks.

**Figure 4 cancers-13-03021-f004:**
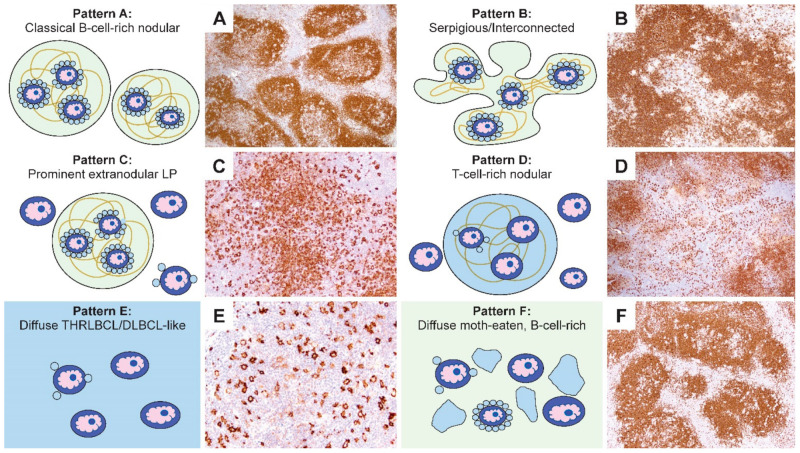
Comparison of the schematic representations and CD20-stained NLPHL patterns. (**A**–**F**) Representative CD20-stained examples of the six NLPHL patterns highlight the number and localization of the LP cells, as well as the content and configuration of small B cells and T cells in the microenvironment in patterns **A**–**F**. [Original magnifications: **A**,**B**,**D**,**F** ×60; **C**,**E** ×150].

**Figure 5 cancers-13-03021-f005:**
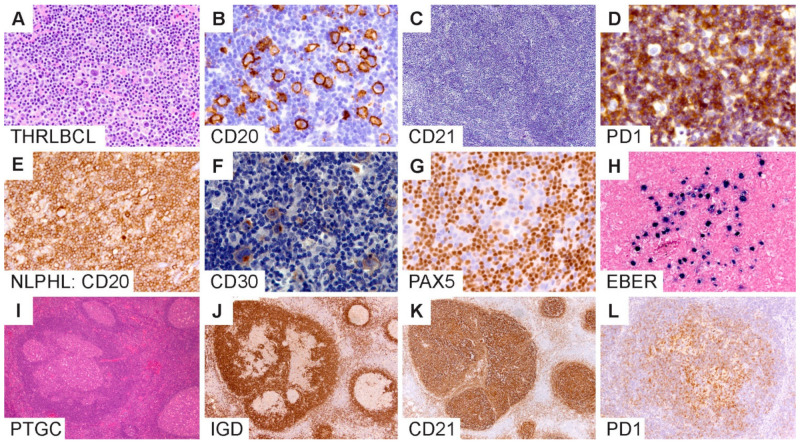
Diagnostic boundaries surrounding NLPHL. (**A**–**D**) T/histiocyte-rich large B-cell lymphomas show diffuse growth, scattered LP cells and an absence of FDC meshworks. (**E**–**H**) NLPHL mimic of lymphocyte-rich classic Hodgkin lymphoma shows the expression of CD20, CD30, dim PAX5 and EBV within LP cells. (**I**–**L**) Progressive transformation of germinal centers show involution of the IgD-positive mantle zone B-cells associated with CD21-positive FDC meshworks and an atypical distribution of PD1-positive T cells in an affected nodule. [Original magnifications: **A**,**C**,**I**,**J**,**K** ×60; **B**,**D**,**E**,**F**,**G**,**H**,**L** ×150].

**Figure 6 cancers-13-03021-f006:**
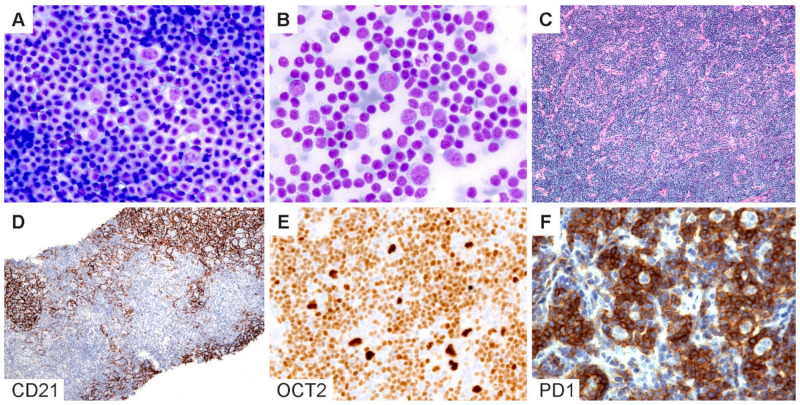
Cytologic diagnosis of NLPHL. (**A**,**B**) Wright–Giemsa-stained touch preparations show LP cells in a background of small lymphoid cells and occasional inflammatory cells. (**C**) H&E section of a core needle biopsy does not readily reveal a nodular architecture. (**D**) A CD21 stain highlights the FDC meshworks and confirms the presence of nodules. (**E**) Bright OCT2 staining and (**F**) prominent PD1-positive rings lend support for the diagnosis of NLPHL. [Original magnifications: **A**,**B** ×600; **C**,**D** ×40; **E**,**F** ×150].

**Table 1 cancers-13-03021-t001:** Immunophenotypic features of NLPHL.

Category	Immunohistologic Marker	Expression	Comments
Pan B-cell markers	CD20, CD79a, CD19	Positive	CD20 is negative in 1–3% and focal/weak in 6% [32].
B-cell transcription factors	PAX5, OCT2, BOB1	Positive	Strong expression in majority; loss or downregulation in a subset of cases.
Immunoglobulins	Kappa/Lambda	Positive	Light chain restriction may be difficult to detect in tissue biopsies.
	J-chain	Positive	
	IgD	Positive subset	Distinct subset of NLPHL [40,41,42].
Pan-leukocyte	CD45RB/LCA	Positive	
Germinal center B-cell markers	BCL6, HGAL, LMO2	Positive	BCL6 FISH positive in NLPHL [43,44].
	CD10	Negative	
Classic Hodgkin lymphoma markers	CD30	Infrequent	Typically weak or subset of LP cells; positive in immunoblasts [27,33,34,35,36].
	CD15	Infrequent	Positive in 2–7% [27,33,34,35,36].
Follicular dendritic cell markers	CD21, CD23	Positive	Loss in diffuse THRLBCL-like NLPHL and THRLBCL [27,45].
T follicular helper T-cell markers	PD1, ICOS	Positive	Form rosettes or rings [46]; loss or downregulation in variant patterns [45].
	CD57	Positive	Variable staining/less sensitive.
Immune checkpoint	PDL1	Positive	Variable staining in LP cells and histiocytes [47,48].
EBV	EBER	Rare	LP cells or bystander B-cells [49,50].
Other/Miscellaneous	MUM1/IRF4	Positive	Strong expression similar to CHL.
	EMA	Positive	Variable staining.
	MEF2B	Positive	Negative in CHL [38].
	STAT6	Negative	Lack nuclear positivity unlike CHL; weak cytoplasmic staining is present in 74% of NLPHL [39]
	IgG4	Negative	Frequent IgG4+ clusters in PTGC [51].

**Table 2 cancers-13-03021-t002:** Recommended immunohistochemistry panels for the diagnosis of NLPHL and differential diagnostic considerations.

Priority/Use	Panel of Markers	Utility
1st tier (Recommended)	CD20, CD3, CD30, CD21 or CD23, PAX5 or OCT2, PD1	Standard panel for initial diagnosis of NLPHL; may vary with biopsy type, practice setting and differential diagnostic considerations.
2nd tier (Desirable)	IgD, EBV (EBER)	Perform if available.
3rd tier (Optional)	CD79a, CD19, BOB1, BCL6, MUM1, CD15, ALK1, Kappa, Lambda	Perform if necessary to confirm or exclude other considerations in the differential diagnosis.
Core needle biopsy for initial diagnosis	CD20, CD3, CD30, PAX5 or OCT2, CD21 or CD23, EBV (EBER); additional B markers, CD15 and ALK1 as necessary	Differential diagnosis includes CHL; if NLPHL is suspected, a surgical biopsy should be performed for definitive diagnosis.
Core needle biopsy in patients with prior NLPHL	OCT2, CD3, CD21 or CD23, IgD, PD1	Primary purpose is to assess for recurrence and/or progression; CD20 may be negative due to prior rituximab therapy. Use another pan-B marker if OCT2 is unavailable.
Separation from classic Hodgkin lymphoma	CD30, CD15, EBER, B-cell transcription factors, IgD, J-chain, MEF2B, STAT6 and BCL6 FISH	NLPHL can mimic lymphocyte-rich CHL. Awareness of aberrant phenotypes should prompt additional workup.
PTCL, AITL and other lymphomas of T follicular helper cells	PD1, ICOS, additional T-cell markers, molecular TCR rearrangement and/or mutational profiling	NLPHL with increased T cells can mimic T cell lymphoma; lack of an aberrant T cell phenotype and TCR rearrangements confirm NLPHL.
CD20-negative NLPHL	Additional B-cell markers, including CD79a, CD19, OCT2, BOB1, MUM1	Awareness of aberrant phenotypes should prompt additional workup.
Progressive transformation of germinal centers	OCT2, IgD, CD21	Presence of PTGC should prompt a search for NLPHL given the known co-occurrence.
EBV+ large atypical cells	Additional B-cell markers, including transcription factors, CD30 and CD15	EBV+ NLPHL is rare; exclude CHL, EBV+ DLBCL and lymphoproliferative disorders arising in infectious or immunodeficiency settings.

## Data Availability

No new data were created or analyzed in this study. Data sharing is not applicable to this article.

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
