# Peer review of "Pitfalls in the Diagnosis of Nodular Lymphocyte Predominant Hodgkin Lymphoma: Variant Patterns, Borderlines and Mimics"

_cancers, 2021, doi:10.3390/cancers13123021_

Round 1
Reviewer 1 Report
The manuscript “Pitfalls in the diagnosis of Nodular Lymphocyte Predominant Hodgkin Lymphoma: variant patterns, borderlines and mimics” by Sheren Younes et coll. is a comprehensive and very detailed review of the current knowledge and diagnostic approach to a rare lymphoma entity. The review focuses on the cytologic/architectural and immunohistochemical features of Nodular Lymphocyte Predominant Hodgkin Lymphoma (NLPHL) and on the criteria useful for differential diagnosis from those lymphomas and reactive entity that may mimic it.
The manuscript is well organized and follows a clear and rational scheme. The literature cited is up to date and absolutely adequate.
Only minor and marginal points, mainly editing errors, need to be addressed.
Page 2, line 75
“TMA” instead of “TME”
Page 5, lines 140-149
is a repetition of the previous paragraph
Figure 5, “labels to figures”
the same letters (A-D) are repeated in all the tumor entities (instead of the E-H and I-J for the second and third respectively).
Is the label “PTCL” in the third row actually “PTGC”?
Author Response
Response to Reviewer 1
The manuscript “Pitfalls in the diagnosis of Nodular Lymphocyte Predominant Hodgkin Lymphoma: variant patterns, borderlines and mimics” by Sheren Younes et coll. is a comprehensive and very detailed review of the current knowledge and diagnostic approach to a rare lymphoma entity. The review focuses on the cytologic/architectural and immunohistochemical features of Nodular Lymphocyte Predominant Hodgkin Lymphoma (NLPHL) and on the criteria useful for differential diagnosis from those lymphomas and reactive entity that may mimic it.
The manuscript is well organized and follows a clear and rational scheme. The literature cited is up to date and absolutely adequate.
Only minor and marginal points, mainly editing errors, need to be addressed.
- Page 2, line 75: “TMA” instead of “TME”
This typographical error has been corrected.
- Page 5, lines 140-149 is a repetition of the previous paragraph
The duplicated paragraph has been deleted.
- Figure 5, “labels to figures”
the same letters (A-D) are repeated in all the tumor entities (instead of the E-H and I-J for the second and third respectively). Is the label “PTCL” in the third row actually “PTGC”?
Thank you very much for catching the errors in the figure labels. They have been corrected.
Reviewer 2 Report
This well written article reviews the challenges in making the diagnosis of nodular lymphocyte-predominant Hodgkin lymphoma given a number differential diagnoses.
There are some points that should be addressed aditionally:
1) page 1, line 44: for reasons of completeness, the two largest analyses on the treatment of NLPHL patients that had been included in prospective studies should be added to the references (stage IA: Eichenauer et al., JCO, 2015; all other stages: Eichenauer et al., JCO, 2020)
2) page 2, first paragraph: for reasons of completeness, the following analysis on transformation of NLPHL into aggressive B-cell lymphoma should be added to the references: Kenderian et al., Blood, 2016
3) page 4, line 107 and page 10, upper paragraph: The correlation between IgD positivity on the LP cells and the presence of pattern C (and E) should be mentioned and discussed (Untanu et al., Pediatr Blood Cancer, 2018)
4) page 10, line 322-338: the recent publication from Paschold et al., Haematologica, 2021 addressing clonal trajectories in NLPHL in the context of histological transformation should be discussed.
Author Response
Reviewer 2
This well written article reviews the challenges in making the diagnosis of nodular lymphocyte-predominant Hodgkin lymphoma given a number differential diagnoses.
There are some points that should be addressed additionally:
1) page 1, line 44: for reasons of completeness, the two largest analyses on the treatment of NLPHL patients that had been included in prospective studies should be added to the references (stage IA: Eichenauer et al., JCO, 2015; all other stages: Eichenauer et al., JCO, 2020)
These references have been added.
2) page 2, first paragraph: for reasons of completeness, the following analysis on transformation of NLPHL into aggressive B-cell lymphoma should be added to the references: Kenderian et al., Blood, 2016
This reference has been added.
3) page 4, line 107 and page 10, upper paragraph: The correlation between IgD positivity on the LP cells and the presence of pattern C (and E) should be mentioned and discussed (Untanu et al., Pediatr Blood Cancer, 2018)
This reference has been added and further discussed in Section 6.
4) page 10, line 322-338: the recent publication from Paschold et al., Haematologica, 2021 addressing clonal trajectories in NLPHL in the context of histological transformation should be discussed.
This reference has been added and further discussed in Section 6.
Reviewer 3 Report
Younes et al. give a nice overview on the pitfalls of NLPHL diagnostics.
I have minor issues:
- Generally, I would recommend to always separately discuss DLBCL and THRBCL. In its current form the paper implies that THRBCL is a subtype of DLBLC, which is not true; it is an entity on its own.
- Lines 27 and 29: there are two "understand" in the sentence; consider a synonym.
- Lines 42-43: in its current form the sentence implies that NLPHL is more or less inevitably linked to multiple recurrences; this is not true, since only 10-15% will ever experience a relapse; please rephrase.
- Line 45: 3-17% is not significant, but simply a proportion of cases; by means of agreement >50% is significant; please reconsider.
- Lines 50-53: I disagree with the formulation. In the usual case, NLPHL can clearly be separated form THRBCL; what the authors describe applies to a minority of cases; please reconsider.
- Line 61: TME has already been spelled and abbreviated; please use only TME.
- Line 71: LP should be spelled out at first mention.
- Lines 92-95: the authors missed MEF2B (Moore et al. Hum Pathol 2017;68:47), which is a novel and very useful diagnostic marker in that consideration. In addition, the missed to cite important papers dealing with CD30 and CD15 in NLPHL (their own Fan-Paper, but also J Hematopathol 2011;4:175, Mod Pathol 2009;22:1006, J Hematopathol 2009;2:211).
- Table 1: please add MEF2B.
- Lines 106-109: though mentioned at the end of the paper, mentioning the frank male predominance and propensity to relapse in the occasion of Moraxella-linked NLPHL already here may be reasonable.
- Lines 128-137 and 140-149: Exactly the same text is copy-pasted here; this may be a technical error; please check.
- Table 2: MEF2B should be incorporated.
- Lines 180-189: The authors may want to read the novelest data on the prognosis of THRBCL in the R-CHOP era and incorporate them in their discussion: Leuk Lymphoma 2020;61:1372.
- I would recommend to put some more emphasis on the IgD, CD21 and PDL1 stainings (and PDL1 FISH) respecting the diagnostic dilemma of de novo THRBCL vs. progression form NLPHL; may be the authors may want to consult: Griffin et al. Blood 2021;137:1353...?
- Line 231: MEF2B may be mentioned here too.
- I would recommend to put some emphasis on the importance of pSTAT6 stainings (nicely detecting STAT6 mutant cHL; no such mutations have been reported in NLPHL) and BCL6 FISH (commonly translocated in NLPHL, but never in cHL) when dealing with the dilemma of cHL vs. NLPHL; may be the authors may want to consult Woldarska et al. Blood 2003;101:706 (already cited in the refs.) and Juskevicius et al. Lab Invest 2018;98:1487...?
- Line 329: please use "κ" and not k in NF-κB.
- Some of the microphotographs are blurred or out of focus, but this may be due to PDF-conversion. Please be sure to publish only highest quality illustrations.
Author Response
Reviewer 3
Younes et al. give a nice overview on the pitfalls of NLPHL diagnostics. I have minor issues:
Generally, I would recommend to always separately discuss DLBCL and THRBCL. In its current form the paper implies that THRBCL is a subtype of DLBLC, which is not true; it is an entity on its own
Thank you for pointing out the lack of clarity. This has been corrected.
2. Lines 27 and 29: there are two "understand" in the sentence; consider a synonym.
This sentence has been rephrased.
3. Lines 42-43: in its current form the sentence implies that NLPHL is more or less inevitably linked to multiple recurrences; this is not true, since only 10-15% will ever experience a relapse; please rephrase.
Thank you for pointing out the lack of clarity. This sentence has been rephrased.
4. Line 45: 3-17% is not significant, but simply a proportion of cases; by means of agreement >50% is significant; please reconsider.
Thank you for pointing out the lack of clarity. This sentence has been rephrased.
5. Lines 50-53: I disagree with the formulation. In the usual case, NLPHL can clearly be separated form THRBCL; what the authors describe applies to a minority of cases; please reconsider.
Thank you for pointing out the lack of clarity. This sentence has been rephrased.
6. Line 61: TME has already been spelled and abbreviated; please use only TME.
Done
7. Line 71: LP should be spelled out at first mention.
Done
8. Lines 92-95: the authors missed MEF2B (Moore et al. Hum Pathol 2017;68:47), which is a novel and very useful diagnostic marker in that consideration. In addition, the missed to cite important papers dealing with CD30 and CD15 in NLPHL (their own Fan-Paper, but also J Hematopathol 2011;4:175, Mod Pathol 2009;22:1006, J Hematopathol 2009;2:211).
These references have been added and discussed.
9. Table 1: please add MEF2B.
Done
10. Lines 106-109: though mentioned at the end of the paper, mentioning the frank male predominance and propensity to relapse in the occasion of Moraxella-linked NLPHL already here may be reasonable.
Done
11. Lines 128-137 and 140-149: Exactly the same text is copy-pasted here; this may be a technical error; please check.
Duplicate text has been deleted.
12. Table 2: MEF2B should be incorporated.
Done
13. Lines 180-189: The authors may want to read the novelest data on the prognosis of THRBCL in the R-CHOP era and incorporate them in their discussion: Leuk Lymphoma 2020;61:1372.
This references has been added and discussed in Section 6.
14. I would recommend to put some more emphasis on the IgD, CD21 and PDL1 stainings (and PDL1 FISH) respecting the diagnostic dilemma of de novo THRBCL vs. progression form NLPHL; may be the authors may want to consult: Griffin et al. Blood 2021;137:1353...?
This references has been added and discussed.
15. Line 231: MEF2B may be mentioned here too.
Done
16. I would recommend to put some emphasis on the importance of pSTAT6 stainings (nicely detecting STAT6 mutant cHL; no such mutations have been reported in NLPHL) and BCL6 FISH (commonly translocated in NLPHL, but never in cHL) when dealing with the dilemma of cHL vs. NLPHL; may be the authors may want to consult Woldarska et al. Blood 2003;101:706 (already cited in the refs.) and Juskevicius et al. Lab Invest 2018;98:1487...?
This section has been expanded according to suggestions and relevant references have been included.
17. Line 329: please use "κ" and not k in NF-κB.
Done
18. Some of the microphotographs are blurred or out of focus, but this may be due to PDF-conversion. Please be sure to publish only highest quality illustrations.
We agree that this is due to PDF conversion. The original submitted images are of very high quality.
Reviewer 4 Report
The review by Younes et al. on the "Pitfalls in the Diagnosis of Nodular Lymphocyte Predominant Hodgkin Lymphoma: Variant Patterns, Borderlines and Mimics", is a very well written review. As a clinician, I found it very compelling and informative. I particularly love table 2 and the pictures provided by the authors, as well as the figures.
I only found some minor spelling errors:
line 37: it says "scare tumor cells", i believe it should be "scarce"
line 75: I believe it should be TME and not TMA
line 104: it should be core, not corre
lines 140-149: are duplicate from lines 128-137
line 174: not sure what "singly" is supposed to mean
line 228: I believe it should be "NLPHL exhibits an intact.." rather than "..and.."
Author Response
Reviewer 4
The review by Younes et al. on the "Pitfalls in the Diagnosis of Nodular Lymphocyte Predominant Hodgkin Lymphoma: Variant Patterns, Borderlines and Mimics", is a very well written review. As a clinician, I found it very compelling and informative. I particularly love table 2 and the pictures provided by the authors, as well as the figures.
I only found some minor spelling errors:
line 37: it says "scare tumor cells", i believe it should be "scarce"
line 75: I believe it should be TME and not TMA
line 104: it should be core, not corre
lines 140-149: are duplicate from lines 128-137
line 174: not sure what "singly" is supposed to mean
line 228: I believe it should be "NLPHL exhibits an intact.." rather than "..and.."
Thank you for the close reading of our manuscript. All points above have been corrected and/or rephrased.